# Regioselective Synthesis of NO-Donor (4-Nitro-1,2,3-triazolyl)furoxans via Eliminative Azide–Olefin Cycloaddition

**DOI:** 10.3390/molecules28196969

**Published:** 2023-10-07

**Authors:** Irina A. Stebletsova, Alexander A. Larin, Ivan V. Ananyev, Leonid L. Fershtat

**Affiliations:** 1N.D. Zelinsky Institute of Organic Chemistry, Russian Academy of Sciences, 47 Leninsky Prospect, 119991 Moscow, Russia; irinastebl@icloud.com (I.A.S.); al_larin@ioc.ac.ru (A.A.L.); 2D.I. Mendeleev University of Chemical Technology of Russia, 9 Miusskaya Square, 125047 Moscow, Russia; 3N.S. Kurnakov Institute of General and Inorganic Chemistry, Russian Academy of Sciences, GSP-1, Leninsky Prospect, 31, 119991 Moscow, Russia; i.ananyev@gmail.com

**Keywords:** nitrogen heterocycles, furoxan, triazole, cycloaddition, NO donors, Griess assay

## Abstract

A facile and efficient method for the regioselective [3 + 2] cycloaddition of 4-azidofuroxans to 1-dimethylamino-2-nitroethylene under *p*-TSA catalysis affording (4-nitro-1,2,3-triazolyl)furoxans was developed. This transformation is believed to proceed via eliminative azide–olefin cycloaddition resulting in its complete regioselectivity. The developed protocol has a broad substrate scope and enables a straightforward assembly of the 4-nitro-1,2,3-triazole motif. Moreover, synthesized (4-nitro-1,2,3-triazolyl)furoxans were found to be capable of NO release in a broad range of concentrations, thus providing a novel platform for future drug design and related biomedical applications of heterocyclic NO donors.

## 1. Introduction

Nitrogen heterocycles are the most frequently occurring structural motifs in various pharmaceuticals and promising drug candidates [1,2,3,4]. According to the U.S. FDA database, >59% of clinically used small-molecule medicines incorporate a nitrogen heterocycle subunit [5,6]. However, the construction of individual pharmaceutical scaffolds using known synthetic methodologies often involves multi-step and energy-consuming procedures or suffers from a lack of reproducibility and scalability. Therefore, the creation of novel step-economy protocols for the assembly of various nitrogen-containing heterocyclic scaffolds remains highly urgent [7,8,9].

Among pharmacologically active substances, heterocyclic NO donors have become an important subclass in organic and medical chemistry [10,11,12,13,14]. In contrast to widely known vasodilator nitroglycerin, heterocyclic NO donors are hydrolytically stable, do not stimulate nitrate tolerance and demonstrate improved pharmacological profiles. Moreover, biomedical applications of simple heterocyclic NO donors along with their hybridization with other pharmacophoric scaffolds were found to be promising for the creation of new medications to overcome the issue of multidrug resistance [15,16,17]. A variety of heterocyclic NO donors (furoxans [18,19,20,21,22], azasydnones [23,24], sydnone imines [25], triazole oxides [26] and pyridazine dioxides [27]) were synthesized and evaluated for their pharmacological potency so far (Figure 1).

In this series, furoxan (1,2,5-oxadiazole 2-oxide) derivatives exhibit a broad range of pharmacological activities including antibacterial [28], antiparasitic [29] and cytotoxic [30,31,32,33] activities. Due to exogenous NO release, furoxans possess promising antiaggregant properties [34,35,36].

In a series of nitrogen heterocycles, 1,2,3-triazoles have become of paramount importance over the past two decades due to their universal application in diverse fields, such as drug development and medicinal chemistry [37,38,39], materials science and polymers [40,41]. In addition, they contribute to organic synthesis by acting as synthetic precursors to many valuable compounds [42,43,44]. In this regard, nitro-1,2,3-triazoles acquire additional significance due to their potential application in energetic materials science and for medicinal chemistry needs [45,46,47]. Taking into account the potency of the molecular hybridization tool in the development of new drug candidates, a method for the synthesis of previously unknown (nitro-1,2,3-triazolyl)furoxans is desired.

Previously, a couple of synthetic approaches toward the construction of a (1,2,3-triazolyl)-1,2,5-oxadiazole biheterocyclic core were proposed. [3 + 2] Cycloaddition of azidofuroxans to acetylenes proceeded under substantially harsh conditions and provided mixtures of regioisomeric 1,2,3-triazoles, while an analogous reaction with 1,3-dicarbonyl compounds had a narrow substrate scope (Figure 1a) [48,49]. Moreover, this approach did not allow the installation of the nitro group onto the 1,2,3-triazole core. [3 + 2] Cycloaddition of 4-amino-3-azidofurazan to 1-nitro-2-morpholinoethylene contributed to the formation of 4-nitro-1,2,3-triazole with a furazan moiety, which, however, is not capable of NO release (Figure 1b) [50]. In addition, this reaction proceeded under prolonged heating in an ionic liquid medium and suffered both from a narrow substrate scope and low yields. These examples clearly demonstrate that the fine tunability of the reactivity of 4-azidofuroxans may be achieved by introducing various additives to achieve both high regioselectivity and a broad scope. Herein, we report on a regioselective, *p*-TSA-catalyzed eliminative azide–olefin cycloaddition of 4-azidofuroxans to 1-dimethylamino-2-nitroethylene for the synthesis of (4-nitro-1,2,3-triazolyl)furoxans (Figure 1c).

## 2. Results and Discussion

Recently, our research group created direct methods for the synthesis of various functionally substituted furoxans. It is also known that the nitro group in 4-nitrofuroxans is prone to nucleophilic displacement due to high electrophilicity of ring carbon atoms [51,52]. In particular, a couple of azidofuroxans were recently prepared upon treatment of 4-nitrofuroxans with NaN_3_ [48]. Using this furoxan reactivity pattern, we prepared a series of starting 4-azidofuroxans **2a**–**v** from the readily available nitro derivatives **1a**–**v** in good and high yields (Figure 2).

4-Azido-3-(*p*-tolyl)furoxan (**2a**) was chosen as a model substrate to optimize the reaction conditions for the synthesis of target (4-nitro-1,2,3-triazolyl)furoxans **3a**–**v** (Table 1). 1-Dimethylamino-2-nitroethylene was used as a convenient dipolarophile in all reactions. Refluxing of substrate **2a** with 1-dimethylamino-2-nitroethylene in various ratios and solvents afforded target 4-nitro-1,2,3-triazole **3a** in low yields (up to 30%, entries 1–8). Interestingly, an addition of Lewis acids did not improve the reaction outcome but resulted in a yield decrease (entries 9–13). Utilization of *m*CPBA as an oxidizer to convert the dimethylamino group to the corresponding *N*-oxide was also inefficient (entry 14). To our delight, more fruitful results were obtained upon *p*-toluenesulfonic acid (*p*-TSA) catalysis (entries 15–18). The optimal amount of *p*-TSA was found to be 15 mol.% (entries 16–18). Interestingly, commercially available pyridinium *p*-toluenesulfonate (PPTS) also catalyzed the studied transformation, albeit the yield of **3a** was lower (entry 19). Therefore, the optimal conditions for the synthesis of (4-nitro-1,2,3-triazolyl)furoxan **3a** were found to be a utilization of 5 equiv. of dipolarophile, 15 mol.% of *p*-TSA in refluxing MeCN (entry 16). It should also be noted, that in all cases, the formation of regioisomeric 5-nitro-1,2,3-triazole **3a′** was not observed.

Having established optimal conditions, we next surveyed the substrate scope of our eliminative azide–olefin cycloaddition by employing an array of azides **2a**–**v** and 1-dimethylamino-2-nitroethylene, and the results are summarized in Figure 3. It was found that azidofuroxans **2c**–**m** bearing an o-, m- or p-substituted phenyl ring underwent cycloaddition smoothly, although nitrotriazoles **3k**,**l** possessing a strongly electron-withdrawing CF_3_ or NO_2_ group at the para-position were formed in lower yields. To our delight, (4-nitro-1,2,3-triazolyl)furoxans **3n**–**q** incorporating di- and trisubstituted phenyl rings were also obtained in good yields. Moreover, a variety of heteroaromatic substituents such as 2-pyridyl-, 6-nitropiperonyl- and 5-nitrofuryl- well tolerated the investigated protocol resulting in a formation of corresponding (4-nitro-1,2,3-triazolyl)furoxans **3r**–**t**. 3-Alkyl-4-azidofuroxans **2u**,**v** underwent eliminative azide–olefin cycloaddition providing target products **3u**,**v** confirming broad applicability of the developed method. It should also be emphasized that in all cases, reaction proceeded regioselectively and the formation of regioisomeric 5-nitro-1,2,3-triazoles was not observed.

All synthesized compounds were characterized by multinuclear (^1^H, ^13^C, ^14^N) NMR spectroscopy, IR spectroscopy, high-resolution mass spectrometry and elemental analysis. The structures of **2a**, **3a** and **3c** were additionally confirmed by X-ray diffraction study (Figure 2).

The molecule of **2a** is nearly planar in crystal: the mean deviation from the mean-square plane composed by non-hydrogen atoms is only 0.032 Å. This conformation is expected owing to the combination of π-donor (4-methylphenyl) and π-acceptor (azidofuroxane) fragments in **2a**. In its turn, the crystal packing of **2a** is of a layer type (Appendix A): planar molecules are bound into layers by the CH…O and CH…N interactions (C…O 3.353(2) and 3.394(2) Å, C…N 3.361(2) Å), whereas the π…π stacking interactions are formed between the layers.

The formal replacement of the azido group with the nitrotriazole fragments in **3a** and **3c** results in the steric repulsion between substituted phenyl rings and triazole moieties. For instance, the C6-C5-C1-C2 and C1-C2-N3-N4 torsion angles in **3a** equal 64.8(2)° and 40.0(2)°, respectively. It is interesting to note that the conjugation of the central furoxan fragment in **3a** with its substituents can be considered as saturable: the C6-C5-C1-C2 and C1-C2-N3-N4 torsion angles are respectively equal to 38.8° and 59.8° in the equilibrium isolated molecule of **3a** modelled at the PBE0-D3/def2TZVP level. In other words, the rotation of one substituent induced by crystal packing effects upon the formal gas-to-crystal transition is totally compensated by the rotation of the other one.

This saturability is further observed in the crystal of **3c**. The bromophenyl substituent in **3c** is disordered over two places with the C6-C5-C1-C2 torsion angles being equal to 118.8(2)° and 96.5(2)°. Owing to this nearly perpendicular arrangement of the phenyl and furoxan cycles, the conjugation within the furoxan–triazole fragment becomes substantial: the C1-C2-N3-N4 torsion angle is only 9.3(2)°. In the isolated molecule of **3c**, the conjugation within the furoxan–triazole fragment again becomes smaller (the C1-C2-N3-N4 torsion angle is 29.1°), whereas the conjugation between the furoxan and bromophenyl moieties becomes larger (the C6-C5-C1-C2 torsion angle is 58.7°).

Nevertheless, despite this significant non-planarity of the molecules of **3a** and **3c**, the layer-type packing motifs are observed in both crystals (Appendix A). The intralayer binding is achieved by a number of the O…π interactions between the furoxane and nitro groups in **3a** (the shortest O…O distances are 2.953(1) and 3.033(1) Å), and by the O…π (the shortest O…X distances are 2.748(2), 3.008(2) and 3.142(3) Å) and the CH…O interactions (C…O 3.203(3)–3.248(3) Å) in **3c**. The hydrophobic contacts are found between the layers in both **3a** and **3c**.

A plausible reaction mechanism of eliminative azide–olefin cycloaddition is outlined in Figure 4. We assume that there are two different ways toward the formation of 1,2,3-triazoline intermediate **A**. Firstly, p-TSA as a Brønsted acid coordinates with the dimethylamino group of the dipolarophile, enabling regioselective [3 + 2] cycloaddition. Thus, generated 1,2,3-triazoline intermediate **A** undergoes elimination of dimethylamine (HNMe_2_) providing 4-nitro-1,2,3-triazole **3** and dimethylamine, which couples with anion **B**, that occurred from p-TSA deprotonation, and forms the salt **C**. Since catalytic amounts of p-TSA are sufficient for the transformation to occur, we propose that dimethylammonium p-toluenesulfonate **C** is also able to catalyze the cycloaddition step, forming a catalytic cycle.

Since furoxans correspond to NO donors, we investigated the ability of the synthesized (4-nitro-1,2,3-triazolyl)furoxans **3** to release NO. The formation of the nitrite anion as a result of the oxidation of NO can be quantified in accordance with the Griess assay and thus can serve as a reliable tool for measuring the NO release. Synthesized (4-nitrotriazolyl)furoxans **3** were subjected to 1 h incubation in the presence of L-cystein under physiological conditions (pH 7.4; 37 °C), and the amount of NO_2_ formed was measured using the spectrophotometric method. It was found that (4-nitrotriazolyl)furoxans containing aromatic or heteroaromatic substituents emit NO fluxes in a broad range of 8.5–72.4%. Interestingly, compounds bearing aliphatic substituents or electron-withdrawing moieties in the aromatic ring (**3j**, **3o**, **3s**) release smaller amounts of NO (8.5–12.1%). It is important to note that furoxanyltriazole **3p** incorporating 3,4-dimethoxyphenyl group demonstrated the highest NO-donor ability (72.4%). Overall, these results might be helpful in the development of novel NO-donor drug candidates with various pharmacological activities (Figure 3).

## 3. Conclusions

In summary, we developed a convenient and straightforward approach to an assembly of (4-nitro-1,2,3-triazolyl)furoxans based on eliminative azide–olefin cycloaddition of 4-azidofuroxans and 1-dimethylamino-2-nitroethylene under *p*-TSA catalysis. The reported method has a number of advantages including a broad substrate scope and complete regioselectivity resulting in the construction of the 4-nitro-1,2,3-triazole motif. Synthesized (4-nitro-1,2,3-triazolyl)furoxans were found to be capable of NO release in a broad range of concentrations under physiological conditions. Therefore, our results contribute to the enlargement of the available libraries of NO-donor substances and unveil novel opportunities in drug design and related biomedical applications.

## 4. Materials and Methods

### 4.1. General Methods


*CAUTION! Although we have encountered no difficulties during preparation and handling of azides **1a**–**w** described in this paper, they are potentially explosive and may be sensitive to impact and friction. Mechanical actions of these species, involving scratching or scraping, must be avoided. Any manipulations must be carried out by using appropriate standard safety precautions.*


All reactions were carried out in well-cleaned, oven-dried glassware with magnetic stirring. ^1^H, ^13^C NMR spectra were recorded on a Bruker AM-300 (300.13 and 75.47 MHz, respectively) spectrometer and referenced to residual solvent peak. ^14^N NMR spectra were measured on a Bruker AM-300 (21.69 MHz) spectrometer using MeNO_2_ (δ^14^N = 0.0 ppm) as an external standard. The chemical shifts are reported in ppm (δ). Mass spectra were measured using a Finnigan MAT INCOS-50 instrument. The IR spectra were recorded on the Simex FT-801 IR-Fourier spectrometer in the 4000–550 cm^−1^ region (spectral resolution 4 cm^−1^) using the universal optical attenuated total reflection (ATR) accessory with ZnSe crystal plate. ZaIR 3.5 software (Simex, Russia) was used to carry out baseline correction and normalization of FEAR spectra. A background (air) measurement was taken for every sample processed. The peaks corresponding to CO_2_ vibrations were removed using the “straight line generation” option in the ZaIR 3.5 software (Simex). Raw spectra were preprocessed using a simple two-point linear subtraction baseline correction method. Two points, 900 and 1850 cm^−1^, were selected outside the wavenumber region of interest that showed no variation across all samples. Spectra were the vector normalized. Spectrum smoothing was not performed. High-resolution mass spectra were recorded on a Bruker microTOF spectrometer with electrospray ionization (ESI). All measurements were performed in a positive (+MS) ion mode (interface capillary voltage: 4500 V) with scan range *m*/*z*: 50–3000. External calibration of the mass spectrometer was performed with Electrospray Calibrant Solution (Fluka). A direct syringe injection was used for all analyzed solutions in MeCN (flow rate: 3 μL min^−1^). Nitrogen was used as nebulizer gas (0.4 bar) and dry gas (4.0 L∙min^−1^); interface temperature was set at 180 °C. All spectra were processed by using Bruker Data Analysis 4.0 software package. Elemental analyses were performed by the CHN Analyzer Perkin-Elmer 2400. Analytical thin-layer chromatography (TLC) was carried out on Merck 25 TLC silica gel 60 F_254_ aluminum sheets. The visualization of the TLC plates was accomplished with a UV light. All standard reagents were purchased from Aldrich or Acros Organics and used without further purification. 4-Azido-3-phenylfuroxan **1b** was obtained according to the previously described procedure [48].

### 4.2. X-ray Crystallography

X-ray diffraction studies were carried out at 100K using the four-circle Rigaku Synergy S diffractometer equipped with a HyPix6000HE area-detector (kappa geometry, shutterless ω-scan technique, monochromatized Cu K_α_-radiation) for **1a** and the Bruker D8 Quest diffractometer equipped with a PhotonIII area-detector (ω- and φ-scan technique, monochromatized Mo K_α_-radiation) for **3a** and **3c**. The intensity data were integrated and corrected for absorption and decay by the CrysAlisPro program for **1a** and by the APEX3 program (SAINT [53], SADABS [54]) for **3a** and **3c.** All structures were solved by dual-space method SHELXT [55] and refined against F^2^ using SHELXL-2018 software (version 2014/6) [56]. All non-hydrogen atoms were refined with individual anisotropic displacement parameters. All hydrogen atoms were found in the difference Fourier synthesis and refined as riding atoms with relative isotropic displacement parameters. A rotating group model was applied for methyl groups. The bromophenyl substituent in the **3c** structure was found to be disordered over two places with population ratio 95:5. All relevant crystal data and refinement details are listed in Appendix A. The CCDC 2290794–2290796 contain all additional information on crystal structures and refinement.

The density functional theory calculations for **3a** and **3c** were performed using the Gaussian program [57] at the PBE0-D3 [58,59,60]/def2TZVP level. Equilibrium structures of both compounds correspond to minimums on the potential energy surface according to the calculations of the Hessian of electronic energy (ultrafine grids, no imaginary modes were found).

### 4.3. Synthetic Procedures

#### 4.3.1. General Procedure for the Synthesis of 4-Azidofuroxans **2a**–**v**

Sodium azide (20.4 mmol, 1.328 g) was added in one portion to a vigorously stirred, ice-cooled solution of the corresponding 4-nitrofuroxan **1** (8.2 mmol) in DMSO (15 mL). Then the reaction mixture was stirred for 3 h at 20 °C until the consumption of substrate **1** (TLC monitoring, eluent CHCl_3_/CCl_4_, 1:1). The resulting mixture was poured onto 30 g of ice and extracted with CH_2_Cl_2_ (3 × 15 mL), and combined organic layers were washed with H_2_O (3 × 15 mL) and dried over MgSO_4_. Filtration of the drying agent and evaporation of the solvent afforded target 4-azidofuroxans **1**.

*4-Azido-3-(*p*-tolyl)furoxan* (**2a**): yield 1.72 g (97%), light yellow solid; mp 100–101 °C, R*_f_* (CH_2_Cl_2_) = 0.85. ^1^H NMR (300 MHz, DMSO-[d_6_]) δ, ppm: 7.85 (d, 2H, *J* = 8.4 Hz, Ar), 7.41 (d, 2H, *J* = 8.4 Hz, Ar), 2.39 (s, 3H, CH_3_); ^13^C NMR (75.5 MHz, CDCl_3_) δ, ppm: 152.7, 141.5, 129.7, 126.5, 118.5, 108.9, 21.6; ^14^N NMR (21.7 MHz, DMSO-[d_6_]): δ = −145.8 (br s, N_3_). IR (KBr), *ν*: 2918, 2148, 1922, 1592, 1518, 1450, 1404, 1331, 1317, 1289, 1238, 1111, 1066, 971, 854, 821, 735 cm^−1^. HRMS (ESI) calcd. for C_9_H_8_N_5_NaO_2_^+^: 240.0492. Found: 240.0485 [M+Na]^+^.

*4-Azido-3-(2-bromophenyl)furoxan* (**2c**): yield 1.75 g (76%), yellow oil; R*_f_* (CHCl_3_/CCl_4_, 4:1) = 0.79. ^1^H NMR (300 MHz, CDCl_3_) δ, ppm: 7.72–7.75 (m, 1H, Ar), 7.36–7.51 (m, 3H, Ar); ^13^C NMR (75.5 MHz, CDCl_3_) δ, ppm: 148.5, 128.9, 128.1, 127.3, 123.2, 119.4, 117.7, 105.2; ^14^N NMR (21.7 MHz, CDCl_3_): δ = −152.3 (s, N_3_). IR (KBr), *ν*: 2144, 1608, 1497, 1438, 1331, 1259, 1133, 1071, 969, 837, 759 cm^−1^. MS (70 eV, *m*/*z* (%)): 225 (1) [M–NO]^+^, 195 (3) [M−2NO]^+^, 281 (4) [M]^+^, 144 (10) [M–Br–N_3_–O]^+^, 114 (19) [M–Br–N_3_–NO_2_]^+^, 30 (100) [NO]^+^. Calcd. for C_8_H_4_BrN_5_O_2_ (%): C, 34.07; H, 1.43; N, 24.83. Found (%): C, 33.89; H, 1.59; N, 24.59.

*4-Azido-3-(2-fluorophenyl)furoxan* (**2d**): yield 1.27 g (70%), yellow solid; mp 122–124 °C, R*_f_* (CH_2_Cl_2_) = 0.93. ^1^H NMR (300 MHz, DMSO-[d_6_]) δ, ppm: 7.66–7.75 (m, 2H, Ar), 7.42–7.51 (m, 2H, Ar); ^13^C NMR (75.5 MHz, DMSO-[d_6_]) δ, ppm: 159.9 (d, *J* = 253.3 Hz), 158.2, 134.6 (d, *J* = 8.5 Hz), 131.7 (d, *J* = 1.9 Hz), 125.6 (d, *J* = 3.5 Hz), 117.0 (d, *J* = 20.0 Hz), 109.1 (d, *J* = 14.2 Hz), 107.1; ^14^N NMR (21.7 MHz, DMSO-[d_6_]): δ = −148.9 (s, N_3_). IR (KBr), *ν*: 2924, 2141, 1805, 1725, 1599, 1550, 1502, 1453, 1404, 1326, 1271, 1092, 970, 846, 753 cm^−1^. Calcd. for C_8_H_4_FN_5_O_2_ (%): C, 43.45; H, 1.82; N, 31.67. Found (%): C, 43.52; H, 1.93; N, 31.50.

*4-Azido-3-(2-nitrophenyl)furoxan* (**2e**): yield 1.60 g (79%), orange solid; mp 100–102 °C, R*_f_* (CH_2_Cl_2_) = 0.88. ^1^H NMR (300 MHz, CDCl_3_) δ, ppm: 8.27 (d, 1H, *J* = 8.1 Hz, Ar), 7.87–7.68 (m, 3H, Ar); ^13^C NMR (75.5 MHz, CDCl_3_) δ, ppm: 152.8, 147.5, 134.1, 132.3, 131.6, 125.8, 116.1, 107.9; ^14^N NMR (21.7 MHz, CDCl_3_): δ = −13.5 (s, NO_2_), δ = −147.8 (s, N_3_). IR (KBr), *ν*: 2976, 2152, 1611, 1553, 1517, 1481, 1383, 1323, 1207, 1128, 1049, 964, 847, 727 cm^−1^. HRMS (ESI) calcd. for C_8_H_8_N_7_O_4_^+^: 266.0625. Found: 266.0632 [M+NH_4_]^+^. HRMS (ESI) calc. for C_8_H_4_N_6_NaO_4_^+^: 271.0181. Found: 271.0186 [M+Na]^+^.

*4-Azido-3-(2-(trifluoromethyl)phenyl)furoxan* (**2f**): yield 1.97 g (89%), white solid; mp 116–118 °C, R*_f_* (CH_2_Cl_2_) = 0.83. ^1^H NMR (300 MHz, CDCl_3_) δ, ppm: 7.86–7.89 (m, 1H, Ar), 7.71–7.78 (m, 2H, Ar), 7.43–7.49 (m, 1H, Ar); ^13^C NMR (75.5 MHz, CDCl_3_) δ, ppm: 153.4, 132.7, 132.3, 131.9, 130.8, 127.5 (q, *J* = 4.6 Hz), 124.9, 119.5 (q, *J* = 92.7 Hz), 108.1; ^14^N NMR (21.7 MHz, CDCl_3_): δ = −147.3 (s, N_3_). IR (KBr), *ν*: 2978, 2148, 1606, 1566, 1484, 1422, 1316, 1273, 1226, 1128, 1051, 967, 844, 786 cm^−1^. HRMS (ESI) calcd. for C_9_H_4_F_3_NaN_5_O_2_^+^: 294.0221. Found: 294.0209 [M+Na]^+^.

*4-Azido-3-(3-bromophenyl)furoxan* (**2g**): yield 1.50 g (65%), beige solid; mp 124–126 °C, R*_f_* (CHCl_3_/CCl_4_, 4:1) = 0.85. ^1^H NMR (300 MHz, CDCl_3_) δ, ppm: 8.22 (t, 1H, *J* = 1.9 Hz, Ar), 8.01–8.05 (m, 1H, Ar), 7.63–7.67 (m, 1H, Ar), 7.41 (t, 1H, *J* = 8.0 Hz, Ar); ^13^C NMR (75.5 MHz, CDCl_3_) δ, ppm: 152.5, 133.9, 130.5, 129.2, 125.0, 123.4, 123.1, 107.8; ^14^N NMR (21.7 MHz, CDCl_3_): δ = −147.4 (s, N_3_). IR (KBr), *ν*: 2922, 2153, 1593, 1485, 1393, 1334, 1272, 1211, 1078, 985, 857, 751 cm^−1^. HRMS (ESI) calcd. for C_8_H_4_^79^BrN_5_NaO_2_^+^: 303.9432. Found: 303.9441 [M+Na]^+^.

*4-Azido-3-(3-chlorophenyl)furoxan* (**2h**): yield 0.60 g (31%), white solid; mp 65–66 °C, R*_f_* (CH_2_Cl_2_) = 0.92. ^1^H NMR (300 MHz, CDCl_3_) δ, ppm: 8.06–8.10 (m, 1H, Ar), 7.95–8.02 (m, 1H, Ar), 7.44–7.51 (m, 2H, Ar); ^13^C NMR (75.5 MHz, CDCl_3_) δ, ppm: 152.5, 135.2, 131.0, 130.3, 126.4, 124.5, 123.2, 107.8; ^14^N NMR (21.7 MHz, CDCl_3_): δ = −146.0 (s, N_3_). IR (KBr), *ν*: 2973, 2923, 2852, 2150, 1642, 1562, 1486, 1395, 1335, 1274, 1220, 1122, 1055, 899, 752 cm^−1^. HRMS (ESI) calcd. for C_8_H_4_ClN_5_NaO_2_^+^: 259.9957 (^35^Cl), 261.9924 (^37^Cl). Found: 259.9946 (^35^Cl), 261.9917 (^37^Cl) [M+Na]^+^.

*4-Azido-3-(3-methoxyphenyl)furoxan* (**2i**): yield 0.99 g (52%), white solid; mp 116–118 °C, R*_f_* (CH_2_Cl_2_) = 0.84. ^1^H NMR (300 MHz, CDCl_3_) δ, ppm: 7.62–7.65 (m, 2H, Ar), 7.41–7.46 (m, 1H, Ar), 7.03–7.07 (m, 1H, Ar), 3.88 (s, 3H, OCH_3_); ^13^C NMR (75.5 MHz, CDCl_3_) δ, ppm: 159.8, 152.8, 130.1, 122.6, 119.0, 116.9, 111.7, 108.8; ^14^N NMR (21.7 MHz, CDCl_3_): δ = −146.3 (s, N_3_). IR (KBr), *ν*: 2947, 2918, 2144, 1631, 1547, 1503, 1481, 1383, 1322, 1274, 1134, 1028, 962, 873, 777 cm^−1^. Calcd. for C_9_H_7_N_5_O_3_ (%): C, 46.36; H, 3.03; N, 30.03. Found (%): C, 46.18; H, 3.09; N, 29.88.

*4-Azido-3-(3-nitrophenyl)furoxan* (**2j**): yield 1.48 g (73%), yellow solid; mp 120–122 °C, R*_f_* (CHCl_3_/CCl_4_, 4:1) = 0.83. ^1^H NMR (300 MHz, CDCl_3_) δ, ppm: 8.95–8.96 (m, 1H, Ar), 8.46–8.49 (m, 1H, Ar), 8.35–8.38 (m, 1H, Ar), 7.73–7.79 (m, 1H, Ar); ^13^C NMR (75.5 MHz, CDCl_3_) δ, ppm: 152.4, 148.5, 131.7, 130.3, 125.3, 123.4, 121.5, 107.5; ^14^N NMR (21.7 MHz, CDCl_3_): δ = −14.3 (s, NO_2_), δ = −148.2 (s, N_3_). IR (KBr), *ν*: 2925, 2150, 1659, 1593, 1530, 1492, 1467, 1350, 1286, 1216, 1080, 999, 876, 794 cm^−1^. HRMS (ESI) calcd. for C_8_H_4_N_6_NaO_4_^+^: 271.0184. Found: 271.0186 [M+Na]^+^.

*4-Azido-3-(4-(trifluoromethyl)phenyl)furoxan* (**2k**): yield 1.31 g (59%), light yellow solid; mp 122–124 °C, R*_f_* (CH_2_Cl_2_) = 0.88. ^1^H NMR (300 MHz, CDCl_3_) δ, ppm: 8.23 (d, 2H, *J* = 8.3 Hz, Ar), 7.79 (d, 2H, *J* = 8.3 Hz, Ar); ^13^C NMR (75.5 MHz, CDCl_3_) δ, ppm: 152.6, 132.5 (d, *J* = 33.0 Hz), 128.3 (d, *J* = 52.0 Hz), 126.9, 125.9 (q, *J* = 3.7 Hz), 125.1, 108.0; ^14^N NMR (21.7 MHz, CDCl_3_): δ = −147.8 (s, N_3_). IR (KBr), *ν*: 2976, 2921, 2152, 1629, 1563, 1514, 1485, 1390, 1323, 1237, 1173, 1054, 955, 846, 771 cm^−1^. Calcd. for C_9_H_4_F_3_N_5_O_2_ (%): C, 39.87; H, 1.49; N, 25.83. Found (%): C, 40.04; H, 1.33; N, 25.69.

*4-Azido-3-(4-nitrophenyl)furoxan* (**2l**): yield 1.34 g (66%), yellow solid; mp 116–118 °C, R_f_ (CH_2_Cl_2_) = 0.78. ^1^H NMR (300 MHz, DMSO-[d_6_]) δ, ppm: 8.44 (d, 2H, *J* = 9.0 Hz, Ar), 8.22 (d, 2H, *J* = 9.0 Hz, Ar); ^13^C NMR (75.5 MHz, DMSO-[d_6_]) δ, ppm: 153.7, 148.6, 128.5, 128.1, 124.6, 109.2; ^14^N NMR (21.7 MHz, DMSO-[d_6_]): δ = −9.3 (s, NO_2_), −146.5 (s, N_3_). IR (KBr): 2924, 2138, 1709, 1598, 1520, 1457, 1404, 1330, 1196, 1065, 971, 847, 752 cm^−1^. Calcd. for C_8_H_4_N_6_O_4_ (%): C, 38.72; H, 1.62; N, 33.87. Found (%): C, 38.85; H, 1.87; N, 33.63.

*4-Azido-3-(4-fluorophenyl)furoxan* (**2m**): yield 1.52 g (84%), light yellow solid; mp 103–105 °C, R*_f_* (CH_2_Cl_2_) = 0.90. ^1^H NMR (300 MHz, CDCl_3_) δ, ppm: 8.07–8.12 (m, 2H, Ar), 7.20–7.28 (m, 2H, Ar); ^13^C NMR (75.5 MHz, CDCl_3_) δ, ppm: 162.4 (d, *J* = 253.7 Hz), 152.5, 128.9 (d, *J* = 8.6 Hz), 117.6 (d, *J* = 3.4 Hz), 116.3 (d, *J* = 22.1 Hz), 108.3; ^14^N NMR (21.7 MHz, CDCl_3_): δ = −146.6 (s, N_3_). IR (KBr), *ν*: 2971, 2150, 1594, 1553, 1476, 1384, 1325, 1231, 1170, 1066, 964, 845, 771 cm^−1^. Calcd. for C_8_H_4_FN_5_O_2_ (%): C, 43.45; H, 1.82; N, 31.67. Found (%): C, 43.60; H, 1.93; N, 31.48.

*4-Azido-3-(2,4-dichlorophenyl)furoxan* (**2n**): yield 1.53 g (69%), light yellow solid; mp 56–58 °C, R*_f_* (CH_2_Cl_2_) = 0.95. ^1^H NMR (300 MHz, CDCl_3_) δ, ppm: 7.59 (d, 1H, *J* = 2.0 Hz, Ar), 7.42–7.45 (m, 1H, Ar), 7.28–7.35 (m, 1H, Ar); ^13^C NMR (75.5 MHz, CDCl_3_) δ, ppm: 153.1, 138.5, 135.7, 132.4, 130.6, 127.9, 118.9, 107.9; ^14^N NMR (21.7 MHz, CDCl_3_): δ = −147.0 (s, N_3_). IR (KBr), *ν*: 2926, 2150, 1725, 1586, 1490, 1370, 1259, 1104, 1052, 955, 855, 777 cm^−1^. Calcd. for C_8_H_3_Cl_2_N_5_O_2_ (%): C, 35.32; H, 1.11; N, 25.74. Found (%): C, 35.14; H, 1.18; N, 25.48.

*4-Azido-3-(3-chloro-4-nitrophenyl)furoxan* (**2o**): yield 1.18 g (51%), orange solid; mp 122–124 °C, R*_f_* (CH_2_Cl_2_) = 0.88. ^1^H NMR (300 MHz, acetone-[d_6_]) δ, ppm: 8.62 (d, 1H, *J* = 2.2 Hz, Ar), 8.37–8.41 (m, 1H, Ar), 7.83 (d, 1H, *J* = 8.7 Hz, Ar); ^13^C NMR (75.5 MHz, acetone-[d_6_]) δ, ppm: 153.1, 140.8, 136.4, 131.4, 123.4, 122.4, 118.7, 107.8; ^14^N NMR (21.7 MHz, acetone-[d_6_]): δ = −14.0 (s, NO_2_), −147.5 (s, N_3_). IR (KBr), *ν*: 2970, 2922, 2117, 1741, 1592, 1529, 1467, 1402, 1337, 1282, 1205, 1066, 997, 906, 860, 829 cm^−1^. Calcd. for C_8_H_3_ClN_6_O_4_ (%): C, 34.00; H, 1.07; N, 29.74. Found (%): C, 34.23; H, 0.93; N, 29.56.

*4-Azido-3-(3,4-dimethoxyphenyl)furoxan* (**2p**): yield 1.76 g (82%), orange solid; mp 131–133 °C, R*_f_* (CH_2_Cl_2_) = 0.90. ^1^H NMR (300 MHz, CDCl_3_) δ, ppm: 7.66–7.69 (m, 2H, Ar), 6.97–7.00 (d, 1H, *J* = 8.7 Hz, Ar), 3.96 (s, 6H, 2xOCH_3_); ^13^C NMR (75.5 MHz, CDCl_3_) δ, ppm: 152.6, 151.0, 149.1, 120.4, 113.7, 111.1, 109.8, 109.0, 56.1, 56.0; ^14^N NMR (21.7 MHz, CDCl_3_): δ = −148.7 (s, N_3_). IR (KBr), *ν*: 2974, 2923, 2851, 2167, 1658, 1612, 1582, 1517, 1483, 1396, 1267, 1217, 1150, 1018, 921, 883, 807 cm^−1^. HRMS (ESI) calc. for C_10_H_10_N_5_O_4_^+^: 264.0727. Found: 264.0730 [M+H]^+^.

*4-Azido-3-(3,4,5-trimethoxyphenyl)furoxan* (**2q**): yield 1.94 g (81%), beige solid; mp 120–121 °C. ^1^H NMR (300 MHz, CDCl_3_) δ, ppm: 8.55 (s, 1H, Ar), 7.19 (s, 1H, Ar), 3.94 (s, 1H, OCH_3_), 3.93 (s, 1H, OCH_3_), 3.90 (s, 1H, OCH_3_); ^13^C NMR (75.5 MHz, CDCl_3_) δ, ppm: 152.3, 150.0, 147.5, 145.9, 144.9, 125.2, 121.0, 104.6, 61.2, 56.1; ^14^N NMR (21.7 MHz, CDCl_3_): δ = −146.9 (s, N_3_). IR (KBr), *ν*: 2975, 2922, 2199, 1680, 1566, 1485, 1396, 1353, 1295, 1243, 1172, 1061, 970, 826, 713 cm^−1^. Calcd. for C_11_H_11_N_5_O_5_ (%): C, 45.06; H, 3.78; N, 23.88. Found (%): C, 44.89; H, 3.90; N, 23.62.

*4-Azido-3-(pyridin-2-yl)furoxan* (**2r**): yield 1.15 g (69%), light yellow solid; mp 81–83 °C, R*_f_* (CH_2_Cl_2_) = 0.91. ^1^H NMR (300 MHz, CDCl_3_) δ, ppm: 8.81 (d, 1H, *J* = 4.9 Hz, Py), 8.22–8.25 (m, 1H, Py), 7.88–7.94 (m, 1H, Py), 7.41–7.45 (m, 1H, Py); ^13^C NMR (75.5 MHz, CDCl_3_) δ, ppm: 153.0, 150.1, 142.7, 137.2, 125.0, 122.4, 118.9; ^14^N NMR (21.7 MHz, CDCl_3_): δ = −146.8 (s, N_3_). IR (KBr), *ν*: 2979, 2135, 1592, 1483, 1418, 1341, 1296, 1213, 1139, 1049, 993, 854 cm^−1^. HRMS (ESI) calc. for C_7_H_5_N_6_O_2_^+^: 205.0468. Found: 205.0474 [M+H]^+^.

*4-Azido-3-(6-nitro-1,3-benzodioxol-5-yl)furoxan* (**2s**): yield 1.72 g (72%), brick solid; mp 166–168 °C, R*_f_* (CH_2_Cl_2_) = 0.75. ^1^H NMR (300 MHz, CDCl_3_) δ, ppm: 7.73 (s, 1H, Ar), 6.99 (s, 1H, Ar), 6.26 (s, 2H, CH_2_); ^13^C NMR (75.5 MHz, CDCl_3_) δ, ppm: 152.9, 152.3, 150.4, 111.4, 109.8, 108.2, 106.7, 104.1; ^14^N NMR (21.7 MHz, CDCl_3_): δ = −12.3 (s, NO_2_), −145.6 (s, N_3_). IR (KBr), *ν*: 2924, 2157, 1612, 1529, 1468, 1418, 1360, 1328, 1263, 1213, 1161, 1026, 919, 883, 783, 743 cm^−1^. HRMS (ESI) calc. for C_9_H_5_N_6_O_6_^+^: 293.0265. Found: 293.0259 [M+H]^+^.

*4-Azido-3-(5-nitrofuran-2-yl)furoxan* (**2t**): yield 0.95 g (49%), beige solid; mp 121–123 °C, R*_f_* (CH_2_Cl_2_) = 0.80. ^1^H NMR (300 MHz, CDCl_3_) δ, ppm: 7.48 (d, 1H, *J* = 3.9 Hz, Furan), 7.39 (d, 1H, *J* = 3.9 Hz, Furan); ^13^C NMR (75.5 MHz, CDCl_3_) δ, ppm: 152.5, 151.0, 139.1, 114.6, 112.5, 103.5; ^14^N NMR (21.7 MHz, CDCl_3_): δ = −31.7 (s, NO_2_), −148.3 (s, N_3_). IR (KBr), *ν*: 2927, 2141, 1725, 1587, 1503, 1459, 1404, 1332, 1246, 1194, 1147, 1087, 1023, 996, 848, 812, 792, 752 cm^−1^. Calcd. for C_6_H_2_N_6_O_5_ (%): C, 30.26; H, 0.85; N, 35.29. Found (%): C, 30.12; H, 0.98; N, 35.03.

*4-Azido-3-methylfuroxan* (**2u**): yield 0.75 g (65%), yellow oil; R*_f_* (CHCl_3_) = 0.88. ^1^H NMR (300 MHz, CDCl_3_) δ, ppm: 2.08 (s, 3H, CH_3_); ^13^C NMR (75.5 MHz, CDCl_3_) δ, ppm: 154.0, 107.3, 6.8. IR (KBr), *ν*: 2930, 2140, 1625, 1508, 1381, 1306, 1237, 1108, 1023, 880, 801 cm^−1^. Calcd. for C_3_H_3_N_5_O_2_ (%): C, 25.54; H, 2.14; N, 49.64. Found (%): C, 25.32; H, 2.31; N, 49.42.

*4-Azido-3-cyclohexylfuroxan* (**2v**): yield 1.62 g (95%), yellow solid; mp 71–73 °C, R*_f_* (CH_2_Cl_2_) = 0.94. ^1^H NMR (300 MHz, CDCl_3_) δ, ppm: 2.55–2.61 (m, 1H, Cy), 1.12–1.75 (m, 10H, Cy); ^13^C NMR (75.5 MHz, CDCl_3_) δ, ppm: 154.2, 113.6, 33.1, 27.6, 25.6, 25.4; ^14^N NMR (21.7 MHz, CDCl_3_): δ = −146.6 (s, N_3_). IR (KBr), *ν*: 2927, 2856, 2142, 1708, 1599, 1496, 1451, 1367, 1306, 1251, 1201, 1048, 952, 883, 823, 761 cm^−1^. Calcd. for C_8_H_11_N_5_O_2_ (%): C, 45.93; H, 5.30; N, 33.48. Found (%): C, 45.64; H, 5.49; N, 33.16.

#### 4.3.2. General Procedure for the Synthesis of 4-(4-Nitro-1H-1,2,3-triazol-1-yl)furoxans **3a**–**v**

*N,N*-Dimethylformamide dimethyl acetal (680 µL, 5.1 mmol) was added to a solution of nitromethane (270 μL, 5 mmol) in MeCN (10 mL). The reaction mixture was stirred for 3 h at 20 °C and volatiles were evaporated on a rotary evaporator affording 1-dimethylamino-2-nitroethylene. Thus obtained dipolarophile was dissolved in anhydrous MeCN (10 mL) followed by the addition of the corresponding 4-azidofuroxan **2** (1 mmol) and *p*-TSA monohydrate (29 mg, 0.15 mmol). The reaction mixture was stirred at 40 °C for 3 h and then refluxed for 69 h. After the disappearance of azides **2** on TLC, the solvent was distilled off under reduced pressure and the target product was purified by column chromatography (eluent CH_2_Cl_2_ or CHCl_3_/CCl_4_, 4:1).

*3-(4-p-Tolyl)-4-(4-nitro-1H-1,2,3-triazol-1-yl)furoxan* (**3a**): yield 161 mg (56%), light yellow solid; mp 132–134 °C, R*_f_* (CH_2_Cl_2_) = 0.7. ^1^H NMR (300 MHz, acetone-[d_6_]) δ, ppm: 9.68 (s, 1H, CH), 7.51 (d, 2H, *J* = 8.1 Hz, Ar), 7.37 (d, 2H, *J* = 8.1 Hz, Ar), 2.41 (s, 3H, CH_3_); ^13^C NMR (75.5 MHz, acetone-[d_6_]) δ, ppm: 154.0, 149.3, 142.1, 129.8, 128.1, 125.9, 117.5, 111.1, 20.6. IR (KBr), *ν*: 2922, 1623, 1549, 1523, 1486, 1389, 1325, 1273, 1137, 1112, 1027, 993, 955, 850, 786, 753 cm^−1^. HRMS (ESI) calcd. for C_11_H_12_N_7_O_4_^+^: 306.0945. Found: 306.0954 [M+NH_4_]^+^.

*4-(4-Nitro-1H-1,2,3-triazol-1-yl)-3-phenylfuroxan* (**3b**): yield 140 mg (51%), white solid; mp 132–134 °C, R*_f_* (CH_2_Cl_2_) = 0.7. ^1^H NMR (300 MHz, acetone-[d_6_]) δ, ppm: 9.69 (s, 1H, CH), 7.56–7.65 (m, 5H, Ar); ^13^C NMR (75.5 MHz, acetone-[d_6_]) δ, ppm: 155.0, 149.7, 131.9, 129.6, 128.7, 126.3, 121.1, 111.6. IR (KBr), *ν*: 2922, 1620, 1551, 1513, 1483, 1388, 1345, 1273, 1238, 1137, 1111, 1075, 1027, 1003, 955, 852, 825, 771 cm^−1^. HRMS (ESI) calcd. for C_10_H_6_N_6_NaO_4_^+^: 297.0343. Found: 297.0346 [M+Na]^+^.

*3-(2-Bromophenyl)-4-(4-nitro-1H-1,2,3-triazol-1-yl)furoxan* (**3c**): yield 144 mg (41%), yellow crystals; mp 122–124 °C, R*_f_* (CHCl_3_/CCl_4_, 4:1) = 0.33. ^1^H NMR (300 MHz, acetone-[d_6_]) δ, ppm: 9.82 (s, 1H, CH), 7.86–7.89 (m, 1H, Ar), 7.81–7.84 (m, 1H, Ar), 7.63–7.66 (m, 2H, Ar); ^13^C NMR (75.5 MHz, acetone-[d_6_]) δ, ppm: 154.3, 149.8, 134.1, 133.8, 133.4, 128.9, 124.5, 124.4, 123.1, 111.3. IR (KBr), *ν*: 2923, 1612, 1554, 1502, 1468, 1386, 1326, 1277, 1182, 1138, 1048, 1026, 950, 864, 825, 771 cm^−1^. HRMS (ESI) calcd. for C_10_H_5_BrN_6_NaO_4_: 374.9448 (^79^Br), 376.9428 (^81^Br). Found: 374.9441 (^79^Br), 376.9423 (^81^Br) [M+Na]^+^.

*3-(2-Fluorophenyl)-4-(4-nitro-1H-1,2,3-triazol-1-yl)furoxan* (**3d**): yield 131 mg (45%), white solid; mp 148–150 °C, R*_f_* (CH_2_Cl_2_) = 0.70. ^1^H NMR (300 MHz, acetone-[d_6_]) δ, ppm: 9.79 (s, 1H, CH), 7.81–7.87 (m, 1H, Ar), 7.70–7.78 (m, 1H, Ar), 7.45–7.50 (m, 1H, Ar), 7.34–7.40 (m, 1H, Ar); ^13^C NMR (75.5 MHz, acetone-[d_6_]) δ, ppm: 160.1 (d, *J* = 240.0 Hz), 153.8, 149.4, 134.4 (d, *J* = 8.6 Hz), 131.2,(d, *J* = 1.4 Hz), 125.3 (d, *J* = 3.6 Hz), 124.5, 116.3 (d, *J* = 21.4 Hz), 109.2 (d, *J* = 14.2 Hz), 107.3. IR (KBr), *ν*: 2924, 2855, 1725, 1710, 1597, 1550, 1502, 1453, 1404, 1326, 1214, 1197, 1066, 1026, 971, 847, 793 cm^−1^. Calcd. for C_10_H_5_FN_6_O_4_ (%): C, 41.11; H, 1.72; N, 28.76. Found (%): C, 41.34; H, 1.64; N, 28.52.

*3-(2-Nitrophenyl)-4-(4-nitro-1H-1,2,3-triazol-1-yl)furoxan* (**3e**): yield 217 mg (68%), yellow solid; mp 121–123 °C, R*_f_* (CH_2_Cl_2_) = 0.70. ^1^H NMR (300 MHz, CDCl_3_) δ, ppm: 9.11 (s, 1H, CH), 8.42–8.45 (m, 1H, Ar), 7.85–7.97 (m, 2H, Ar) 7.74–7.79 (m, 1H, Ar); ^13^C NMR (75.5 MHz, CDCl_3_) δ, ppm: 153.5, 148.7, 147.3, 135.0, 133.3, 132.6, 126.2, 121.2, 116.5, 108.5. IR (KBr), *ν*: 2922, 1620, 1553, 1485, 1388, 1322, 1170, 1133, 1067, 1027, 953, 824, 754 cm^−1^. HRMS (ESI) calcd. for C_10_H_5_N_7_NaO_6_^+^: 342.0194. Found: 342.0192 [M+Na]^+^.

*4-(4-Nitro-1H-1,2,3-triazol-1-yl)-3-(2-(trifluoromethyl)phenyl)furoxan* (**3f**): yield 123 mg (36%), light yellow solid; mp 100–102 °C, R*_f_* (CH_2_Cl_2_) = 0.62. ^1^H NMR (300 MHz, CDCl_3_) δ, ppm: 9.02 (s, 1H, CH), 7.79–7.89 (m, 3H, Ar), 7.64–7.68 (m, 1H, Ar); ^13^C NMR (75.5 MHz, CDCl_3_) δ, ppm: 153.5, 148.7, 133.1, 132.7, 132.5, 130.6, 127.4 (q, *J* = 4.7 Hz), 124.9, 121.5, 118.6 (d, *J* = 2.0 Hz), 108.0. IR (KBr), *ν*: 2923, 2851, 1620, 1585, 1553, 1485, 1439, 1389, 1319, 1172, 1121, 1033, 951, 867, 824, 771 cm^−1^. Calcd. for C_11_H_5_F_3_N_6_O_4_ (%): C, 38.61; H, 1.47; N, 24.56. Found (%): C, 38.47; H, 1.64; N, 24.30.

*3-(3-Bromophenyl)-4-(4-nitro-1H-1,2,3-triazol-1-yl)furoxan* (**3g**): yield 208 mg (59%), yellow solid; mp 110–112 °C, R*_f_* (CH_2_Cl_2_) = 0.65. ^1^H NMR (300 MHz, CDCl_3_) δ, ppm: 8.97 (s, 1H, CH), 7.83 (t, 1H, *J* = 1.8 Hz, Ar), 7.70–7.74 (m, 1H, Ar), 7.36–7.50 (m, 2H, Ar); ^13^C NMR (75.5 MHz, CDCl_3_) δ, ppm: 156.3, 153.3, 135.1, 131.3, 130.8, 127.1, 123.4, 123.0, 121.6, 109.2. IR (KBr), *ν*: 2921, 1610, 1529, 1427, 1383, 1351, 1325, 1271, 1231, 1179, 1133, 1075, 1026, 960, 852, 826, 786 cm^−1^. HRMS (ESI) calcd. for C_10_H_5_^79^BrN_6_NaO_4_: 374.9448. Found: 374.9456 [M+Na]^+^.

*3-(3-Chlorophenyl)-4-(4-nitro-1H-1,2,3-triazol-1-yl)furoxan* (**3h**): yield 117 mg (38%), light yellow solid; mp 1118–120 °C, R*_f_* (CH_2_Cl_2_) = 0.54. ^1^H NMR (300 MHz, CDCl_3_) δ, ppm: 8.96 (s, 1H, CH), 7.78 (t, 1H, *J* = 1.9 Hz, Ar), 7.56–7.63 (m, 2H, Ar), 7.49 (t, 1H, *J* = 7.9 Hz, Ar); ^13^C NMR (75.5 MHz, CDCl_3_) δ, ppm: 144.4, 135.5, 132.2, 130.6, 129.3, 127.2, 124.0, 123.8, 116.6. IR (KBr), *ν*: 1620, 1547, 1504, 1477, 1387, 1322, 1302, 1278, 1176, 1135, 1066, 1025, 952, 840 cm^−1^. Calcd. for C_10_H_5_ClN_6_O_4_ (%): C, 38.92; H, 1.63; N, 27.23. Found (%): C, 39.09; H, 1.51; N, 26.99.

*3-(3-Methoxyphenyl)-4-(4-nitro-1H-1,2,3-triazol-1-yl)furoxan* (**3i**): yield 142 mg (47%), white solid; mp 116–118 °C, R*_f_* (CH_2_Cl_2_) = 0.61. ^1^H NMR (300 MHz, CDCl_3_) δ, ppm: 8.90 (s, 1H, CH), 7.41 (t, 1H, *J* = 8.1 Hz, Ar), 7.16 (t, 1H, *J* = 2.1 Hz, Ar), 7.06–7.10 (m, 1H, Ar), 7.00–7.04 (m, 1H, Ar), 3.83 (s, 3H, OCH_3_); ^13^C NMR (75.5 MHz, CDCl_3_) δ, ppm: 160.1, 148.2, 130.6, 123.6, 120.5, 120.2, 117.8, 113.6, 110.2, 55.5. IR (KBr), *ν*: 2918, 2846, 1631, 1603, 1548, 1503, 1430, 1383, 1323, 1274, 1112, 1047, 1004, 977, 824, 784 cm^−1^. Calcd. for C_11_H_8_N_6_O_5_ (%): C, 43.43; H, 2.65; N, 27.63. Found (%): C, 43.29; H, 2.76; N, 27.41.

*3-(3-Nitrophenyl)-4-(4-nitro-1H-1,2,3-triazol-1-yl)furoxan* (**3j**): yield 163 mg (51%), light yellow solid; mp 121–123 °C, R*_f_* (CH_2_Cl_2_) = 0.86. ^1^H NMR (300 MHz, acetone-[d_6_]) δ, ppm: 9.44 (s, 1H, CH), 8.52 (t, 1H, *J* = 2.0 Hz, Ar), 8.35–8.38 (m, 1H, Ar), 7.91–7.95 (m, 1H, Ar), 7.72–7.77 (m, 1H, Ar); ^13^C NMR (75.5 MHz, acetone-[d_6_]+CDCl_3_) δ, ppm: 154.0, 148.5, 148.4, 134.5, 130.6, 126.1, 124.6, 123.8, 122.3, 109.6. IR (KBr), *ν*: 2921, 1626, 1551, 1502, 1440, 1383, 1326, 1291, 1170, 1142, 1091, 1023, 960, 902, 844, 808 cm^−1^. Calcd. for C_10_H_5_N_7_O_6_ (%): C, 37.63; H, 1.58; N, 30.72. Found (%): C, 37.82; H, 1.42; N, 30.48.

*4-(4-Nitro-1H-1,2,3-triazol-1-yl)-3-(4-(trifluoromethyl)phenyl)furoxan* (**3k**): yield 112 mg (33%), light yellow solid; mp 145–147 °C, R*_f_* (CH_2_Cl_2_) = 0.84. ^1^H NMR (300 MHz, CDCl_3_) δ, ppm: 9.01 (s, 1H, CH), 7.75–7.84 (m, 4H, Ar); ^13^C NMR (75.5 MHz, CDCl_3_) δ, ppm: 148.0, 133.4, 129.8, 129.1, 126.4 (q, *J* = 3.6 Hz, CF_3_), 125.0, 123.8, 123.4, 123.1, 121.4, 109.5. IR (KBr), *ν*: 2922, 1620, 1551, 1519, 1486, 1388, 1322, 1164, 1111, 1066, 1027, 953, 823, 754 cm^−1^. HRMS (ESI) calcd. for C_11_H_5_F_3_N_6_NaO_4_^+^: 365.0217. Found: 365.0207 [M+Na]^+^.

*3-(4-Nitrophenyl)-4-(4-nitro-1H-1,2,3-triazol-1-yl)furoxan* (**3l**): yield 150 mg (47%), pale orange solid; mp 119–121 °C, R*_f_* (CH_2_Cl_2_) = 0.55. ^1^H NMR (300 MHz, acetone-[d_6_]+CDCl_3_) δ, ppm: 9.77 (s, 1H, CH), 8.38–8.44 (m, 2H, Ar), 7.96–8.00 (m, 2H, Ar); ^13^C NMR (75.5 MHz, acetone-[d_6_]+CDCl_3_) δ, ppm: 149.2, 130.1, 127.8, 127.1, 125.6, 124.1, 123.43, 110.4. IR (KBr), *ν*: 1602, 1553, 1517, 1390, 1325, 1113, 1031, 987, 950, 854, 752 cm^−1^. Calcd. for C_10_H_5_N_7_O_6_ (%): C, 37.63; H, 1.58; N, 30.72. Found (%): C, 37.47; H, 1.67; N, 30.49.

*3-(4-Fluorophenyl)-4-(4-nitro-1H-1,2,3-triazol-1-yl)-furoxan* (**3m**): yield 146 mg (50%), light yellow solid; mp 120–122 °C, R*_f_* (CH_2_Cl_2_) = 0.74. ^1^H NMR (300 MHz, CDCl_3_) δ, ppm: 8.97 (s, 1H, CH), 7.61–7.65 (m, 2H, Ar), 8.25–8.29 (m, 2H, Ar); ^13^C NMR (75.5 MHz, CDCl_3_) δ, ppm: 164.4 (d, *J* = 255.5 Hz), 148.2, 130.8 (d, *J* = 9.0 Hz), 123.2, 117.0 (d, *J* = 22.5 Hz), 115.6 (d, *J* = 3.6 Hz), 109.8, 105.8. IR (KBr), *ν*: 2974, 2923, 1627, 1551, 1516, 1486, 1392, 1355, 1324, 1238, 1203, 1112, 1016, 954, 839, 752 cm^−1^. Calcd. for C_10_H_5_FN_6_O_4_ (%): C, 41.11; H, 1.72; N, 28.76. Found (%): C, 40.95; H, 1.90; N, 28.49.

*3-(2,4-Dichlorophenyl)-4-(4-nitro-1H-1,2,3-triazol-1-yl)furoxan* (**3n**): yield 212 mg (62%), pale yellow solid; mp 117–119 °C, R*_f_* (CH_2_Cl_2_) = 0.84. ^1^H NMR (300 MHz, CDCl_3_) δ, ppm: 9.06 (s, 1H, CH), 7.56–7.61 (m, 2H, Ar), 7.50–7.53 (m, 1H, Ar); ^13^C NMR (75.5 MHz, CDCl_3_) δ, ppm: 153.7, 148.6, 139.4, 135.5, 132.7, 130.5, 128.5, 121.5, 118.4, 108.0. IR (KBr), *ν*: 2923, 2853, 1726, 1628, 1551, 1510, 1474, 1407, 1322, 1277, 1180, 1137, 1053, 950, 884, 788 cm^−1^. Calcd. for C_10_H_4_Cl_2_N_6_O_4_ (%): C, 35.01; H, 1.18; N, 24.50. Found (%): C, 34.83; H, 1.30; N, 24.32.

*3-(4-Chloro-3-nitrophenyl)-4-(4-nitro-1H-1,2,3-triazol-1-yl)furoxan* (**3o**): yield 191 mg (54%), yellow solid; mp 134–136 °C, R*_f_* (CH_2_Cl_2_) = 0.45. ^1^H NMR (300 MHz, acetone-[d_6_]) δ, ppm: 9.10 (s, 1H, CH), 8.06 (br. s, 1H, Ar), 7.69 (br. s, 1H, Ar), 7.28 (s, 1H, Ar); ^13^C NMR (75.5 MHz, acetone-[d_6_]) δ, ppm: 156.5, 149.1, 149.0, 146.6, 138.0, 125.4, 117.7, 113.7, 104.9. IR (KBr), *ν*: 2978, 2901, 1613, 1592, 1553, 1480, 1384, 1278, 1080, 1023, 993, 908, 805, 725 cm^−1^. Calcd. for C_10_H_4_ClN_7_O_6_ (%): C, 33.96; H, 1.14; N, 27.73. Found (%): C, 34.13; H, 1.01; N, 27.51.

*3-(3,4-Dimethoxyphenyl)-4-(4-nitro-1H-1,2,3-triazol-1-yl)furoxan* (**3p**): yield 167 mg (50%), white solid; mp 115–117 °C, R*_f_* (CH_2_Cl_2_) = 0.86. ^1^H NMR (300 MHz, CDCl_3_) δ, ppm: 8.92 (s, 1H, CH), 7.20 (d, 1H, *J* = 2.0 Hz, Ar), 7.02 (dd, 1H, *J* = 8.5, 2.0 Hz, Ar), 6.92–6.94 (d, 1H, *J* = 8.5 Hz, Ar), 3.92 (s, 3H, OCH_3_), 3.85 (s, 3H, OCH_3_); ^13^C NMR (75.5 MHz, CDCl_3_) δ, ppm: 153.7, 151.9, 149.5, 148.3, 123.9, 121.7, 111.5, 111.1, 110.5, 110.4, 56.2, 56.1. IR (KBr), *ν*: 2921, 1580, 1517, 1484, 1433, 1387, 1324, 1221, 1149, 1115, 1012, 953, 860, 769 cm^−1^. Calcd. for C_12_H_10_N_6_O_6_ (%): C, 43.12; H, 3.02; N, 25.14. Found (%): C, 42.95; H, 3.15; N, 24.92.

*4-(4-Nitro-1H-1,2,3-triazol-1-yl)-3-(3,4,5-trimethoxyphenyl)furoxan* (**3q**): yield 240 mg (66%), white solid; mp 138–140 °C, R*_f_* (CHCl_3_/EtOAc, 15:1) = 0.38. ^1^H NMR (300 MHz, CDCl_3_) δ, ppm: 8.56 (s, 1H, CH), 7.27 (s, 2H, Ar), 3.91–3.95 (m, 9H, 3 OCH_3_); ^13^C NMR (75.5 MHz, CDCl_3_) δ, ppm: 152.3, 150.0, 147.7, 145.4, 127.7, 124.2, 121.5, 111.5, 105.0, 61.3, 61.2, 56.3. IR (KBr), *ν*: 2924, 1591, 1565, 1488, 1426, 1352, 1296, 1201, 1110, 1061, 1013, 923, 825, 763 cm^−1^. Calcd. for C_13_H_12_N_6_O_7_ (%): C, 42.86; H, 3.32; N, 23.07. Found (%): C, 43.09; H, 3.16; N, 22.88.

*4-(4-Nitro-1H-1,2,3-triazol-1-yl)-3-(pyridin-2-yl)furoxan* (**3r**): yield 137 mg (50%), pale yellow solid; mp 139–141 °C, R*_f_* (CH_2_Cl_2_) = 0.86. ^1^H NMR (300 MHz, CDCl_3_) δ, ppm: 9.13 (s, 1H, CH), 8.51–8.54 (m, 1H, Py), 8.25–8.29 (m, 1H, Py), 7.94–8.00 (m, 1H, Py), 7.43–7.48 (m, 1H, Py); ^13^C NMR (75.5 MHz, CDCl_3_) δ, ppm: 150.2, 148.1, 141.2, 137.8, 125.9, 125.7, 122.5, 110.0. IR (KBr), *ν*: 2922, 1625, 1544, 1466, 1389, 1302, 1272, 1157, 1091, 1033, 989, 866, 785 cm^−1^. HRMS (ESI) calcd. for C_9_H_5_N_7_NaO_4_^+^: 298.0295. Found: 298.0306 [M+Na]^+^.

*3-(6-Nitro-1,3-benzodioxol-5-yl)-4-(4-nitro-1H-1,2,3-triazol-1-yl)furoxan* (**3s**): yield 260 mg (72%), brown solid; mp 138–140 °C, R*_f_* (CH_2_Cl_2_) = 0.86. ^1^H NMR (300 MHz, CDCl_3_) δ, ppm: 9.08 (s, 1H, CH), 7.84 (s, 1H, Ar), 7.06 (s, 1H, Ar), 6.32 (s, 2H, CH_2_); ^13^C NMR (75.5 MHz, CDCl_3_) δ, ppm: 153.1, 151.2, 137.1, 131.6, 121.2, 118.5, 110.5, 110.2, 106.9, 104.5. IR (KBr), *ν*: 2923, 1712, 1625, 1588, 1503, 1481, 1391, 1331, 1265, 1110, 1024, 972, 881, 822 cm^−1^. HRMS (ESI) calcd. for C_11_H_5_N_7_NaO_8_^+^: 386.0081. Found: 386.0092 [M+Na]^+^.

*3-(5-Nitrofuran)-4-(4-nitro-1H-1,2,3-triazol-1-yl)furoxan* (**3t**): yield 152 mg (52%), white solid; mp 139–141 °C, R*_f_* (CH_2_Cl_2_) = 0.50. ^1^H NMR (300 MHz, CDCl_3_) δ, ppm: 9.85 (s, 1H, CH Triazole), 7.77 (d, 1H, *J* = 4.0 Hz, CH Furan), 7.60 (d, 1H, *J* = 4.0 Hz, CH Furan); ^13^C NMR (75.5 MHz, CDCl_3_) δ, ppm: 155.4, 153.9, 146.9, 137.9, 126.9, 116.2, 112.7, 105.9. IR (KBr), *ν*: 2925, 1796, 1725, 1630, 1503, 1475, 1394, 1310, 1244, 1155, 1025, 912, 856, 755 cm^−1^. Calcd. for C_8_H_3_N_7_O_7_ (%): C, 31.08; H, 0.98; N, 31.72. Found (%): C, 30.89; H, 1.11; N, 31.56.

*3-Metyl-4-(4-nitro-1H-1,2,3-triazol-1-yl)furoxan* (**3u**): yield 127 mg (60%), pale yellow solid; mp 129–131 °C, R*_f_* (CH_2_Cl_2_) = 0.45. ^1^H NMR (300 MHz, CDCl_3_) δ, ppm: 9.10 (s, 1H, CH), 2.63 (s, 3H, CH_3_); ^13^C NMR (75.5 MHz, CDCl_3_) δ, ppm: 153.7, 149.8, 121.0, 107.5, 9.3. IR (KBr), *ν*: 2961, 2924, 2854, 1725, 1658, 1592, 1501, 1461, 1388, 1313, 1261, 1179, 1026, 950, 823, 752 cm^−1^. Calcd. for C_5_H_4_N_6_O_4_ (%): C, 28.31; H, 1.90; N, 39.62. Found (%): C, 28.09; H, 1.98; N, 39.39.

*3-(Cyclohexyl)-4-(4-nitro-1H-1,2,3-triazol-1-yl)furoxan* (**3v**): yield 187 mg (67%), beige solid; mp 121–123 °C, R*_f_* (CH_2_Cl_2_) = 0.76. ^1^H NMR (300 MHz, CDCl_3_) δ, ppm: 9.03 (s, 1H, CH), 3.17–3.24 (m, 1H), 1.25–1.96 (m, 10H, Cy); ^13^C NMR (75.5 MHz, CDCl_3_) δ, ppm: 153.7, 148.8, 122.2, 113.4, 33.4, 26.4, 25.6, 25.0. IR (KBr), *ν*: 2929, 2854, 1726, 1608, 1547, 1491, 1449, 1387, 1313, 1270, 1180, 1105, 1049, 987, 881, 823 cm^−1^. Calcd. for C_10_H_12_N_6_O_4_ (%): C, 42.86; H, 4.32; N, 29.99. Found (%): C, 43.09; H, 4.18; N, 29.72.

#### 4.3.3. NO Release Assay

The test molecule (0.1 mmol) was dissolved in DMSO (50 mL). A 20 µL aliquot of the resulting solution was diluted with a phosphate buffer solution (180 µL, Ph 7.4). The final concentration of the tested compound was 2 × 10^−4^ M. The mixture was incubated at 37 °C for 1 h. A 50 µL aliquot of the Griess reagent (prepared by mixing sulfanilamide (4 g), *N*-naphthylethylenediamine dihydrochloride (0.2 g) and 85% H_3_PO_4_ (10 mL) in distilled and deionized water (final volume 100 mL)) was added and incubated for 10 min at 37 °C. UV absorbance at 540 nm was measured using a Multiskan GO Microplate Photometer and calibrated using a standard curve prepared from standard solutions of NaNO_2_ to give the nitrite concentration. All measurements were made in triplicate.

## Data Availability

Data obtained in this project are contained within this article and are available upon request from the authors.

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
