# Peer review of "Regioselective Synthesis of NO-Donor (4-Nitro-1,2,3-triazolyl)furoxans via Eliminative Azide–Olefin Cycloaddition"

_molecules, 2023, doi:10.3390/molecules28196969_

Round 1
Reviewer 1 Report
The paper describes the regioselective synthesis of NO-donor (4-Nitro-1,2,3-triazolyl)furoxans via eliminative azide-Olefin Cycloaddition.
The work is interesting and the presentation good. The compounds prepared are novel and they are fully characterised. The supporting information file is of good quality.
Based on that I suggest that the paper is accepted after the following are addressed.
Corrections/comments:
- Scheme 1: Please add the equivalents of reagents and conditions.
- Table 1: Why was 15 mol% of acid used and not less?
Author Response
The authors are grateful to the reviewer for their valuable comments on our manuscript. We did all our best to improve the manuscript and the responses are provided below.
The paper describes the regioselective synthesis of NO-donor (4-Nitro-1,2,3-triazolyl)furoxans via eliminative azide-Olefin Cycloaddition.
The work is interesting and the presentation good. The compounds prepared are novel and they are fully characterised. The supporting information file is of good quality. Based on that I suggest that the paper is accepted after the following are addressed.
Thank you for your positive opinion about our work.
Corrections/comments:
- Scheme 1: Please add the equivalents of reagents and conditions.
Corrected.
- Table 1: Why was 15 mol% of acid used and not less?
As a rule, Brønsted acid is used in catalytic amounts to promote [3+2]-cycloaddition reactions. We conducted additional optimization studies and found that 15 mol.% is a sufficient amount of p-TSA (see Table 1).
Reviewer 2 Report
This paper describes efficient synthetic methods for drug-like furoxans. Although the mechanistic concept is similar to previous work (Scheme 1b), the yield and substrate scope have improved. The compound's characterization is thorough, and the procedure appears to be easily reproducible. I recommend accepting this paper for publication in Molecules after minor revisions, based on the following comments:
- In Table 1, a control experiment in which the reaction is conducted under the same conditions as entry 15, except without using p-TSA, is required.
- On page 7, please carefully review the compound numbering. In Sentences 174 and 175, compounds 5j, 5o, 5t, and 5p are mentioned, which seems to be incorrect. Figure 2 displays data for compound 5w, which also appears to be incorrect. Moreover, the text mentions, "It is important to note that furoxanyltriazole 5p incorporating 3,4,5-trimethoxyphenyl group demonstrated…". If you mistakenly referred to 3p as 5p, please note that compound 3p has a dimethoxyphenyl group, not a 3,4,5-trimethoxyphenyl group.
- In Scheme 4, this reaction releases dimethylamine as a co-product, which must neutralize a catalytic amount of p-TSA. Please explain why a catalytic amount of p-TSA is sufficient. What is the yield when using 30 mol% of p-TSA? Please address this in the text. Additionally, consider the possibility that p-TSA dimethylamine salt might also be a capable catalyst for this reaction. If so, how about using para-toluenesulfonic acid pyridinium salt (PPTS), a commercially available mild acid, instead of p-TSA? If possible, it is worth trying because the reaction conditions may become milder than the current ones.
- One clear drawback of this method is that it only allows for the synthesis of 4-nitro derivatives. Is there any method to convert this nitro group into other functional groups to enhance the versatility of this methodology? For example, is it possible to perform an aromatic nucleophilic substitution reaction (SNAr reaction) to convert the nitro group into an alkoxy or alkylsulfanyl group, or simply reduce the nitro group to an amino group or hydrogen?
Author Response
The authors are grateful to the reviewer for their valuable comments on our manuscript. We did all our best to improve the manuscript and the responses are provided below.
This paper describes efficient synthetic methods for drug-like furoxans. Although the mechanistic concept is similar to previous work (Scheme 1b), the yield and substrate scope have improved. The compound's characterization is thorough, and the procedure appears to be easily reproducible. I recommend accepting this paper for publication in Molecules after minor revisions, based on the following comments:
Thank you for the helpful advices and positive feedback about the work.
In Table 1, a control experiment in which the reaction is conducted under the same conditions as entry 15, except without using p-TSA, is required.
We tested the reaction conditions without using p-TSA and the reaction did not occur. The corresponding row has been added to Table 1.
On page 7, please carefully review the compound numbering. In Sentences 174 and 175, compounds 5j, 5o, 5t, and 5p are mentioned, which seems to be incorrect. Figure 2 displays data for compound 5w, which also appears to be incorrect. Moreover, the text mentions, "It is important to note that furoxanyltriazole 5p incorporating 3,4,5-trimethoxyphenyl group demonstrated…". If you mistakenly referred to 3p as 5p, please note that compound 3p has a dimethoxyphenyl group, not a 3,4,5-trimethoxyphenyl group.
In the original version of the manuscript we made some numbering misprints. Now all typos were corrected.
In Scheme 4, this reaction releases dimethylamine as a co-product, which must neutralize a catalytic amount of p-TSA. Please explain why a catalytic amount of p-TSA is sufficient. What is the yield when using 30 mol% of p-TSA? Please address this in the text. Additionally, consider the possibility that p-TSA dimethylamine salt might also be a capable catalyst for this reaction. If so, how about using para-toluenesulfonic acid pyridinium salt (PPTS), a commercially available mild acid, instead of p-TSA? If possible, it is worth trying because the reaction conditions may become milder than the current ones.
Thank you for this suggestion. We conducted the studied reaction using PPTS as a catalyst and model product 3a was also formed, albeit in a lower yield than under standard conditions. This data was added to Table 1. Since PPTS was also able to catalyze the formation of nitro-1,2,3-triazoles, we should assume that dimethylammonium p-toluenesulfonate might as well catalyze this process. Therefore, we revised the reaction mechanism and included discussion regarding catalysis with dimethylammonium p-toluenesulfonate.
One clear drawback of this method is that it only allows for the synthesis of 4-nitro derivatives. Is there any method to convert this nitro group into other functional groups to enhance the versatility of this methodology? For example, is it possible to perform an aromatic nucleophilic substitution reaction (SNAr reaction) to convert the nitro group into an alkoxy or alkylsulfanyl group, or simply reduce the nitro group to an amino group or hydrogen?
Indeed, this methods enables a preparation of only 4-nitro-1,2,3-triazoles. At the same time, it is known that nitro group in such derivatives can be reduced to the corresponding amines (see, for example, J. Med. Chem., 2006, 49, 4409-4424; DOI: 10.1021/jm060133g). We might as well study the reactivity of the synthesized compounds and we would like to report these results in due course.
Reviewer 3 Report
The manuscript reports the synthesis of (4-nitro-1,2,3-triazolyl)furoxans in decent yields (and some exceptions with low yields) using a convenient methodology. The compounds are well characterized, and results well-presented and discussed. The manuscript deserves publication in Molecules, upon minor corrections described below.
Introduction
- Line 37: The sentence lists examples of NO-donors such as furoxans, azasydnones, sydnone imines, triazole oxides, pyridazine dioxides. It would be useful if the authors would provide a picture with these structural scaffolds.
Results and discussion
- Line 80: There is a typo in the sentence, it should be Scheme 2 instead of Scheme 1. In the same sentence, “… derivatives 2”, should be described as “… derivatives 2a-w”, for consistency purposes.
In Scheme 2, the compounds should be re-numbered. The nitro compounds should be 1a-w and the 4-azidofuroxans should be 2a-w, and not the opposite as depicted in Scheme 2.
Author Response
The authors are grateful to the reviewer for their valuable comments on our manuscript. We did all our best to improve the manuscript and the responses are provided below.
The manuscript reports the synthesis of (4-nitro-1,2,3-triazolyl)furoxans in decent yields (and some exceptions with low yields) using a convenient methodology. The compounds are well characterized, and results well-presented and discussed. The manuscript deserves publication in Molecules, upon minor corrections described below.
Thank you for the positive feedback about our work.
Introduction
- Line 37: The sentence lists examples of NO-donors such as furoxans, azasydnones, sydnone imines, triazole oxides, pyridazine dioxides. It would be useful if the authors would provide a picture with these structural scaffolds.
Added.
Results and discussion
- Line 80: There is a typo in the sentence, it should be Scheme 2 instead of Scheme 1. In the same sentence, “… derivatives 2”, should be described as “… derivatives 2a-w”, for consistency purposes.
Corrected
In Scheme 2, the compounds should be re-numbered. The nitro compounds should be 1a-w and the 4-azidofuroxans should be 2a-w, and not the opposite as depicted in Scheme 2.
Corrected
Round 2
Reviewer 2 Report
I am satisfied with the revised version. Thank you for considering my comments. In the revised Table1, entry "15" is still in bold. Instead, entry "16" should be in bold in the revised one. Please revise it before publication.